# Timing of Complementary Feeding, Growth, and Risk of Non-Communicable Diseases: Systematic Review and Meta-Analysis

**DOI:** 10.3390/nu14030702

**Published:** 2022-02-08

**Authors:** Maria Carmen Verga, Immacolata Scotese, Marcello Bergamini, Giovanni Simeone, Barbara Cuomo, Giuseppe D’Antonio, Iride Dello Iacono, Giuseppe Di Mauro, Lucia Leonardi, Vito Leonardo Miniello, Filomena Palma, Giovanna Tezza, Andrea Vania, Margherita Caroli

**Affiliations:** 1ASL Salerno, 84019 Vietri Sul Mare, Salerno, Italy; 2ASL Salerno, 84022 Campagna, Salerno, Italy; scotese.ped@libero.it; 3AUSL Ferrara, 44121 Ferrara, Italy; marcelloberga54@gmail.com; 4ASL Brindisi, 72023 Mesagne, Brindisi, Italy; giovanni.simeone@gmail.com; 5Department of Pediatrics, Belcolle Hospital, 01010 Viterbo, Italy; cuomoba@gmail.com; 6Independent Researcher, 84100 Salerno, Italy; gdantonio32@gmail.com; 7Independent Researcher, 82100 Benevento, Italy; iridedello@gmail.com; 8ASL Caserta, Aversa, 81031 Caserta, Italy; presidenza@sipps.it; 9Maternal Infantile and Urological Sciences Department, Sapienza University, 00161 Rome, Italy; lucialeonardi@yahoo.it; 10Nutrition Unit, Department of Pediatrics, “Giovanni XXIII” Children Hospital, “Aldo Moro” University of Bari, 70126 Bari, Italy; vito.miniello@libero.it; 11ASL Salerno, 84091 Battipaglia, Salerno, Italy; menapalma3@gmail.com; 12F. Tappeiner Hospital, 39012 Merano, Bolzano, Italy; giovanna.tezza@gmail.com; 13Independent Researcher, 00162 Rome, Italy; andrea.vania57@gmail.com; 14Independent Researcher, Francavilla Fontana, 72021 Brindisi, Italy; margheritacaroli53@gmail.com

**Keywords:** complementary feeding, early nutrition, human milk, breastfeeding, formula feeding, weaning, growth, overweight, obesity, type 2 diabetes mellitus (DM2), hypertension

## Abstract

No consensus currently exists on the appropriate age for the introduction of complementary feeding (CF). In this paper, a systematic review is conducted that investigates the effects of starting CF in breastfed and formula-fed infants at 4, 4–6, or 6 months of age (i) on growth at 12 months of age, (ii) on the development of overweight/obesity at 3–6 years of age, (iii) on iron status, and (iv) on the risk of developing (later in life) type 2 diabetes mellitus (DM2) and hypertension. An extensive literature search identified seven studies that evaluated the effects of the introduction of CF at the ages in question. No statistically significant differences related to the age at which CF is started were observed in breastfed or formula-fed infants in terms of the following: iron status, weight, length, and body mass index Z-scores (zBMI) at 12 months, and development of overweight/obesity at 3 years. No studies were found specifically focused on the age range for CF introduction and risk of DM2 and hypertension. Introducing CF before 6 months in healthy term-born infants living in developed countries is essentially useless, as human milk (HM) and formulas are nutritionally adequate up to 6 months of age.

## 1. Introduction

Over the past two decades, our growing understanding of the nutritional, immunological, and neurodevelopmental benefits of breastfeeding has led many national and international associations and scientific societies to revise their recommendations on the age at which complementary feeding (CF) should be added to the diet of healthy infants.

The World Health Organization (WHO), which considers exclusive breastfeeding (EBF) crucial not only for nutritional considerations, but also as a public health issue, has changed its previous recommendation on EBF for 4–6 months to recommend it for the first 6 months of life, and then up to 2 years of age or more, with the addition of nutritionally appropriate and safe complementary foods [1].

Later on, the European Commission also formally requested the European Food Safety Authority (EFSA) Panel on Nutrition, Novel Foods and Food Allergens (NDA) to update its 2009 recommendations on the appropriate age for the introduction of CF to define whether the wording “from four months”, already displayed on baby food products according to European directives, should be changed to “from six months” according to the latest WHO recommendations. In its latest document, EFSA brought its scientific opinion on the duration of breastfeeding into line with WHO recommendations, but it did not give specific recommendations on the age at which CF should start [2].

Unfortunately, to date, several factors (personal, socio-economic, cultural, and also related to healthcare services) still influence the decision to replace/supplement human milk (HM) with formula or complementary foods at an early stage [3]. Among these factors, the following elements are likely to play a role:–Lack of support to breastfeeding by healthcare professionals;–Use of growth curves based on formula-fed infants or cultural beliefs about child growth, which can lead to breastfed infants being considered underweight and to supplementation with additional foods to achieve what is wrongly thought to be an appropriate weight gain;–Belief that supplementation is an acceptable routine practice, not an intervention;–Early maternal return to work and unavailability of workplace breastfeeding facilities;–Social disapproval of breastfeeding in public.

In addition to these elements, there is also a lack of perception, on the part of parents, but also of many healthcare workers, that the early addition of liquids or foods other than human milk (HM) or, in case of unavailability of HU in the first 6 months of life, of formula may have negative effects on the health of the infant and child later in life.

Finally, observational studies and surveys have consistently reported an association between early introduction of complementary foods and shorter breastfeeding duration [3].

Even so, no consensus currently exists on the appropriate age for the introduction of CF in relation not only to the assessments of the infant’s psychomotor development and nutritional requirements, but also in terms of potential short- and long-term health outcomes, including iron deficiency and obesity.

Since 2005, WHO has recommended that both breastfed and formula-fed infants should start CF at 6 completed months, i.e., not less than 180 days of life [4].

On the other hand, the current ESPGHAN and AAP recommendations suggest continuing breastfeeding up to 6 months of age, but complementary foods should be introduced to all infants between 17 and 26 weeks, partially contradicting the aforementioned recommendation [5,6].

Several studies have shown that (i) after 6 months of age infants, especially breastfed ones, can be at risk of iron deficiency due to depletion of their iron stores [7], and (ii) the start of CF before 4 months of age is neither necessary nor feasible for all infants because HM and formulas meet all nutritional needs, and because infants younger than 4 months have generally not yet acquired adequate motor skills for spoon-feeding [2].

The recommendations on the appropriate age for the introduction of CF are, therefore, quite conflicting as some uncertainties remain concerning the potential benefits or risks linked to the introduction of CF between 4 and 6 months of age.

### Objectives

The objective of this systematic review (SR) is to provide answers to the following questions:–Does starting CF between 4 and 6 months of age lead to different short-term and long-term nutritional and metabolic outcomes compared with exclusive breastfeeding up to 6 months?–Does starting CF between 4 and 6 months of age lead to different short-term and long-term nutritional and metabolic outcomes compared with exclusive formula or mixed feeding (human milk and baby formula) up to 6 months?

In details, the aims are:To compare the effects on growth at 12 months and the development of overweight/obesity at 3–6 years of EBF for less than 6 months and introduction of CF between 4 and 6 months, compared with EBF for 6 months with the introduction of CF at 6 months of age;To compare the effects on growth at 12 months and the development of overweight/obesity at 3–6 years of exclusive formula feeding (EFF) or mixed for less than 6 months and introduction of CF between 4 and 6 months, compared with EFF for 6 months with the introduction of CF at 6 months.

According to UNICEF [8], in children aged 0 to 59 months, overweight is defined as weight-for-height above +2 SD (standard deviation) of the WHO Child Growth Standards median for children of the same height and sex, while severe overweight (obesity) is when weight-for-height ratio is above +3 SD. In children and young people aged 5–19 years, overweight is defined by a BMI-for-age above 1 SD of the WHO Growth Reference median for children of the same age and sex; severe overweight (above +2 SD) is referred to as obesity, and a BMI-for-age above +3 SD is referred to as severe obesity.

The secondary objective is to compare the effects of both EBF and EFF/mixed, with the introduction of CF between 4 and 6 months, with the effects of the same infant feeding modalities for 6 months, and with the introduction of CF at 6 months of age (180 days), on the child’s iron status at 6–12 months, as well as on the development of DM2 and hypertension later in life [8].

The questions, structured according to PICO, are reported in Appendix A.

## 2. Materials and Methods

Details of the protocol for this SR were registered on PROSPERO and can be accessed at www.crd.york.ac.uk/PROSPERO/display_record.asp?ID=CRD42021273191 (accessed on 12 September 2021)—Registration number: CRD42021273191.

The formulation of clinical questions, the search, the analysis of documents and scientific evidence with specific tools (AGREE II for GLs, AMSTAR 2 for SRs, Cochrane Assessment of risk of bias for RCTs, Newcastle Ottawa Scales for observational studies), and the GRADE method have been previously described [9].

### 2.1. Design of the Studies Included

The sources covered in the review included: government publications, systematic reviews, meta-analyses, randomized controlled trials (RCTs), multicenter studies, observational studies including cohort studies, longitudinal studies, case–control studies, and cross-sectional studies.

Both RCTs and observational studies were included because the effect of introducing CF in the age range “4 to 6 months” compared to 6 months could be reliably assessed both as an experimental intervention and as an exposure factor, while also taking into account potential confounders.

### 2.2. Population

Healthy term-born babies with normal birth weight, breastfed and/or formula-fed, aged 4–24 months or older for long-term outcomes, living in Western industrialized countries were included in this study.

### 2.3. Intervention(s), Exposure(s)

The introduction of CF between 4 and 6 months of age was analyzed.

### 2.4. Comparator(s)/Control

Children from the end of their 6th month to the end of their 7th month of life, were used as comparators.

In addition to the studies that compared the two above-mentioned age groups, studies were also included that did not refer to a specific age, but where no association between age of introduction of CF and outcomes (e.g., growth or obesity risk) had been found by regression methods. This was conducted because, if there is no relationship between age of introduction of CF (independent variable) and outcomes (dependent variables), the result obtained can apply to all ages, including those of the PICOs (structured questions: population, intervention, comparator, outcome) of this SR, hence these studies could be used to provide an answer to the research questions.

### 2.5. Exclusion Criteria

The studies based on populations with characteristics that differed from those established in the PICOs were excluded.

In addition, studies with CF introduction at different ages (e.g., <4 months) or where just a generic age was indicated (e.g., ≥6 months) for CF introduction were excluded along with studies where only liquid foods other than HM and formulas were considered.

Studies with methodological biases likely to affect the confidence in the results obtained were also excluded [10,11].

### 2.6. Main Outcome(s)

–General growth parameters: weight (W), length (L), W/L Z-score ratio, body mass index (BMI); BMI Z-score (zBMI);–Risk of non-communicable diseases (NCDs; overweight/obesity, diabetes mellitus type 2 (DM2), and hypertension).

### 2.7. Additional Outcome(s)

Iron status (hemoglobin, ferritin, serum iron), risk of DM2, and risk of hypertension were also analyzed.

### 2.8. Keywords and Search Strategy

See Appendix A.

### 2.9. Measures of Effect

The standard method of the Cochrane Review Group to synthesize the data was used and the effects were expressed as risk ratio (RR) and risk difference (RD) with 95% confidence intervals (Cis) for categorical data. For continuous outcomes, the mean difference (MD) or standardized main difference (SDM) and the 95% confidence intervals (Cis) were used [9].

### 2.10. Studies Selection. Risk of Bias (Quality) Assessment. Missing Data. List of the Studies Excluded with Relevant Reasons. Data Extraction (Selection and Coding)

See Appendix A.

The selection process and the assessment of the comprehensive methodological quality and of that of the individual studies, including handling missing data and data extraction, were carried out by at least two authors [9].

The SRs and the studies were evaluated using specific tools, (AMSTAR 2 for SRs, Cochrane Assessment of Risk of Bias for RCTs, and Newcastle Ottawa Scales for observational studies), as previously described [9].

The following data were extracted from the studies included: author, year of publication, study design, objective of the study, country, sample, healthy or pathological condition, age, type of intervention, period of follow-up, results, main conclusions of the study, and financing.

### 2.11. Assessment of Heterogeneity

The chi-square test was used to test heterogeneity; to quantify the percentage of total variation across studies due to heterogeneity, the I_2_ index was calculated. The fixed-effect model was used for meta-analysis when enrolled infants and interventions were similar and no significant heterogeneity was found. The sources of heterogeneity were explored by performing subgroup analysis.

### 2.12. Strategy for Data Synthesis

A meta-analysis was conducted when data merging was possible.

In the presence of significant statistical heterogeneity, data were merged using the random effect model. Mantel–Haenszel analysis was used for the dichotomous scores and inverse variance for the continuous scores.

### 2.13. Reporting Bias Assessment

When possible (number of studies included in the meta-analysis ≥10), the outcome reporting bias and the publication bias were assessed by examining the degree of asymmetry of a funnel plot.

### 2.14. Softwares

The Review Manager 5 (RevMan 5) 5.4.1 software [12] was used for the methodological quality of the RCTs, the meta-analyses, and the relevant figures.

The GRADEpro GDT software, developed by the GRADE Working Group, was used to grade the overall quality of evidence and the relevant tables [13].

## 3. Results

The searches of evidence (Appendix A) are summarized in Figure 1, Figure 2 and Figure 3 and the results are given in Appendix A. The meta-analyses are shown in Appendix A, whereas the evaluation of the overall quality of evidence is given in Table 1 and Table 2 and in Appendix A.

Five “Evidence Syntheses” were selected: one Italian consensus statement [14], three SRs of moderate methodological quality [2,15,16], and one high-quality SR [3].

From the five “Evidence Syntheses”, four RCTs were selected [17,18,19,20]. A total of 2 of these [17,18] are the same study conducted on 100 children and 1 reports follow-up data at 18 and 36 months [19]. Only the 18-month follow-up data were included in the final analysis, because the dropout rate at 36 months was 23%.

In addition, two observational studies of moderate quality were selected [21,22].

Further search and selection of studies resulted in the selection of 27 eligible studies, including a good-quality case-control study [23].

No growth data and risk of overweight/obesity at 6 years are included in this SR, because the dropout rate of the studies at this age was above 20%.

As far as the risk of DM2 and hypertension is concerned, no studies were found that specifically focused on the age range for CF introduction considered in this SR.

The list of excluded studies and reasons for exclusion is given in Appendix A.

### 3.1. Exclusively Breastfed (EBF) Infant

#### 3.1.1. Growth at 6 and 12 Months

The results of an RCT on a population of 100 infants (17) on several outcome indicators (presented as Z-scores), including W, L, BMI, W, and H gain at 6 months, confirm that the introduction of CF at 4 or 6 months does not result in statistically significant differences in any of the parameters in question.

A cross-sectional observational study on 571 term infants (CBGS Study) [22] explored whether the introduction of CF at different ages between 3 and 6 months of age, (146 infants [25.6%] at 4.0–4.9 months, 226 infants [39.6%] at 5.0–5.9 months, or the remaining 155 [27.1%] at 6.0–6.9 months) promoted faster growth during infancy. Depending on the type of milk feeding, the study design was as follows:A total of 165 EBF infants were weaned at 4.0–4.9 and 5.0–5.9 months;A total of 86 EBF infants were weaned at 6.0–6.9 months;A total of 103 EFF/mixed infants were weaned at 4.0–4.9 and 5.0–5.9 months;A total of 69 EFF/mixed infants were weaned at 6.0–6.9 months.

No statistically significant difference was observed in W, L, and BMI Z-scores at 12 months related to the age at CF starting.

The overall quality of evidence is moderate.

#### 3.1.2. Overweight/Obesity at 3 Years of Age

An RCT (19) reports the mean difference in zBMI at 18 months in a population of 94 EBF children, with 46 of them being weaned at 6 months and 48 between 4 and 6 months (MD (95% CI) = −0.01 (−0.39–0.37); *p* = 0.95; RR of overweight/obesity at 18 months (95% CI) = 1.30 (0.37–4.56); *p* = 0.68).

A cohort study (21) reports obesity data at 3 years in a group of EBF children who received CF between 4 and 6 months (*n* = 427) or at 6 months (*n* = 98). The results were not statistically significantly different between the 2 subgroups, and no difference in odds of developing overweight/obesity at 3 years (RR = 0.80; 95%IC = 0.51–1.23) was observed.

These results were also confirmed by a more recent case–control study [23] on a sample of 463 children, with 28 (6.1%) of them being overweight/obese. The study in question evaluated the effect of some exposure factors, including the duration of EBF and the age of introduction of fruit and cereal porridge, on the development of overweight/obesity at 3 years of age. The ages of the introduction of fruit and cereal porridge were (median, in parentheses the range: minimum–maximum) 5 (1–13) and 6 (1–24) months, respectively. Linear regression analysis did not show any statistically significant correlation with either of these exposure factors: β coefficient = 0.020 (*p* = 0.743) and 0.011 (*p* = 0.828), respectively.

The quality of evidence is moderate.

#### 3.1.3. Iron status

As for the iron status outcome, the SR of the EFSA Panel on Nutrition, Novel Foods and Food Allergens (2) provides a literature update, as of May 2019. There is only one RCT conducted on a population of infants in a developed country (Iceland) split into 2 groups, one with continued EBF until 6 months, and the other with the introduction of solid food at 4 months while continuing breastfeeding. In both groups, the hemoglobin, mean corpuscular volume, total iron binding capacity, and red cell distribution width were not significantly different. Although significantly higher levels of serum ferritin were observed in the group with the introduction of solid food at 4 months compared to 6 months (*p* = 0.02), serum ferritin was within the normal range in both groups [17].

The overall quality of evidence is moderate (Table 1).

**Table 1 nutrients-14-00702-t001:** Summary of findings for the main comparisons. Blood iron status.

	**Research string: (Introduction CF at 4–6 months) than (introduction CF at 6 months) in order to (blood iron level at 6 months)**					
	Patient or population: (Blood iron level at 6 months) Setting: Outpatient Intervention: (Introduction CF at 4–6 months) Comparator: (Introduction CF at 6 months)					
**Outcomes**	**Anticipated absolute outcome * (95% CI)**		**Relative outcome** **(95% CI)**	**№ of participants** **(studies)**	**Certainty of evidence** **(GRADE)**	**Comments**
	**Risk with (Introduction CF at 4–6 months)**	**Risk with (Introduction CF at 6 months)**				
Serum Hb (Hb) evaluated with: gr/L follow-up: 6 months	The average serum hb was = 0.2	MD = 0.2 (2.44 inferior to 2.48 major)	-	100 (1 RCT) [17].	⨁⨁⨁◯ MODERATE ^a^	
Serum ferritin (SF) evaluated with: ug/L follow-up: 6 months	The average serum ferritin was = 26	MD = 26 (0.1 inferior to 52.1 major)	-	100 (1 RCT) [17].	⨁⨁⨁◯ MODERATE ^a^	

* The risk in the intervention group (and its confidence interval (CI) at 95%) is based on the risk assumed in the control group and on the relative outcome of the intervention (and its CI at 95%). CI: Confidence interval; MD: Mean difference. Explications: “Anticipated absolute outcome” and its subs in bold since they represent the main outcome. ^a^ Large 95% IC.

### 3.2. Exclusively/Predominantly Formula-Fed Infant

#### 3.2.1. Growth at 6 and 12 Months

An RCT [20] was conducted on a population of 41 infants fed exclusively with a formula for 4 months and randomized to receive CF between 16 and 26 weeks of age (first group), or to be EFF and receive CF at 26 weeks of age. The primary outcomes were bone mineral content, serum Ca, P, Mg, calcitonin, and parathormone at 6 months. The secondary outcomes were accounted for by the anthropometric data collected by blinded healthcare professionals. The differences observed in the weight–height gain at 26 weeks were not statistically significant, but as this is only 1 study on a small population sample, its results cannot be considered conclusive.

The overall quality of evidence is moderate.

#### 3.2.2. Overweight/Obesity at 3 Years of Age

A cohort study [21] showed no significant differences in 3 sample populations of formula-fed infants (<4 months, 4–5 months, ≥6 months) between the group with CF introduction between 4–5 months and the group weaned at ≥6 months (RR (95%IC) for developing overweight/obesity at 3 years = 1.24 (0.66–2.33); *p* = 0.50).

The overall quality of evidence is low.

The evaluation of the overall quality of evidence for the main comparisons on growth and risk of overweight/obesity is given in Table 2.

**Table 2 nutrients-14-00702-t002:** Summary of findings for the main comparisons. Growth and risk of overweight/obesity.

Research string: GROWTH						
(Introduction CF at 4–6 months) than (introduction CF at 6 months) in order to (ensure adequate growth at 6–12–18–24 months)						
Patient or population: (Ensure adequate growth at 6–12–18–24 months) Setting: Outpatient Intervention: (Introduction CF at 4–6 months) Comparator: (Introduction CF at 6 months)						
Outcomes	Anticipated absolute outcome * (95% CI)		Relative outcome (95% CI)	№ of participants (studies)	Certainty of evidences (GRADE)	Comments
	Risk with (introduction CF at 4–6 months)	Risk with (introduction CF at 6 months)				
Weight gain Z-score (WGZ) follow-up: average 6 months	The average weight gain Z-score was = −0.01	MD = −0.01 (0.15 inferior to 0.13 major)	-	141 (2 RCT) [17,20]. ^a^	⨁⨁⨁◯ MODERATE ^b^	
Length gain Z-score (LGZ) follow-up: 6 months	The average length gain Z-score was = −0.01	MD = −0.01 (0.21 inferior to 0.19 major)	-	141 (2 RCT) [17,20]. ^a^	⨁⨁⨁◯ MODERATE ^b^	
Weight Z-score (WZ) follow-up: 12 months	N of patients introduction 4–6 n = 372. WZS at 12 months = 0.58 (0.99)–0.39 (0.95). N of patients introduction at 6 months = 155. WZ at 12 months = 0.25 (0.92) *p* = 0.01. Association with CF introduction age, fixed for age, sex, maternal age, parity, deprivation score, milk feeding at 3 months, and growth at precedent cut point (Model 3) = 0.01 (−0.06 to 0.07), *p* = 0.88			527 (1 observational study) [22].	⨁⨁⨁◯ MODERATE	
Length Z-score (LZ) follow-up: 12 months	Patient CF introduction 4–6 months. LZS at 12 months = 0.48 (1.05), 0.23 (1.04). Patient CF introduction at 6 months. LZS at 12 months = 0.00 (1.04) *p* < 0.01. Association with CF introduction age fixed for confounding factors (Model 3) 0.04 (−0.01 to 0.11) *p* = 0.20			(1 observational study) [22].	⨁⨁⨁◯ MODERATE	
BMI Z-score (BZ) follow-up: 12 months	Introduction CF 4–6 months. BMIZ at 12 months 0.42 (0.94)–0.36 (0.83). CF introduction 6 months. BMIZ at 12 months 0.33 (0.84) *p* = 0.33. Association with CF introduction age fixed for confounding factors (Model 3) −0.02 (−0.08 to 0.05) *p* = 0.64			(1 observational study) [22].	⨁⨁⨁◯ MODERATE	
Research string: [RISK OF OVERWEIGHT/OBESITY]						
(Introduction CF at 4–6 months) than (introduction CF at 6 months) in order to develop overweight/obesity at 3–6 years						
Patient or population: Development of overweight/obesity at 3–6 years Setting: Outpatient Intervention: (Introduction CF at 4–6 months) Comparator: (Introduction CF at 6 months)						
Outcomes	Anticipated absolute outcome * (95% CI)		Relative outcome (95% CI)	№ of participants (Studies)	Certainty of evidence (GRADE)	Comments
	Risk with (Introduction CF at 6 months)	Risk with (introduction CF at 4–6 months)				
Overweight/obesity at 18 months (S/O 24–36) evaluated with: BMI Z-score follow-up: 18 months	109 per 1.000	141 per 1.000 (40 at 496)	RR 1.30 (0.37 at 4.56)	94 (1 RCT) [17].	⨁⨁⨁◯ MODERATE ^a^	
Overweight/obesity at 3 years (S/O 6a) evaluated with: RR (95% IC) follow-up: 3 years	2. A total of 463 children, of which 28 (6.1%) were in an overweight/obesity condition. The linear regression analysis did not show a statistically significant correlation with the age of introduction of fruit and cereals: coefficient β, respectively = 0.020 (*p* = 0.743) and 0.011 (*p* = 0.828). 3. Starting CF at 4–6 months (*n* = 427) or at 6 months (*n* = 98). There is no difference in the probability of developing overweight/obesity at 3 years (RR = 0.80; 95%IC = 0.51–1.23)			525 (2 observational studies) [20,22].	⨁⨁◯◯ LOW ^b^	

* The risk in the intervention group (and its confidence interval (CI) at 95%) is based on the risk assumed in the control group and on the relative outcome of the intervention (and its CI at 95%). CI: Confidence interval; MD: Mean difference. Explications: “Anticipated absolute outcome” and its subs in bold since they represent the main outcome. ^a^ A total of 2 publications, but from 1 same study. ^b^ For every exposition factor (BF o FF) the study is unique and with low sample numerosity.

## 4. Discussion

This SR was conducted to support the recommendations on the appropriate age range for the introduction of CF into the diets of healthy term babies with normal birth weight, breastfed and/or formula-fed, ages 4–24 months or older for long term outcomes, living in Western industrialized countries [9].

In GLs and other position papers, as well as in the latest SRs conducted with improved methodological quality, even though 3 out of 4 of these papers have not been updated since 2016, (3, 15, 16), the references provided for this specific question are presented by very few studies: 2 RCTs with a small sample size (17, 20) and 2 observational studies (21–22).

In this SR, evidence search, updated from May 2015 to August 2021, resulted in 27 eligible studies, with only 1 being included and adding to the 5 studies already selected by the SRs. As for the other 26 studies, the most frequent reasons for exclusion were: (i) being set in low-income countries or countries with non-Western dietary styles, (ii) age(s) at weaning not relevant to the question of this SR, and (iii) poor methodological quality due to important loss to follow-up.

Western industrialized countries alone were considered in this analysis, for several reasons: first of all, and in spite of differences even inside each industrialized country due to different socio-economic status among families, the average possibilities in these countries are higher (or even much higher) than those available to families living in low-income areas of our world, hence the different offers to weaning infants are not comparable. Secondly, cultural (and culturally-related feeding practices) are used to differ more between high- and low-income countries (LICs) than among high-income countries alone, hence—once again—a comparison is quite difficult. Finally, a small weight for the gestational age at birth, which is very frequent in developing countries, constitutes a risk factor for the development of NCDs in later ages and therefore this would have been a further confounding factor for understanding the role of CF in the development of long-term NCDs. For these reasons, the results of the studies conducted in the LICs are difficult to transfer to our children. As for the age at the start of CF, few studies compared the introduction of CF at 4–6 months with the introduction of CF at 6 months of age; despite a great deal of evidence already available, even in recent years, research efforts continued to explore the effects of early introduction (<4 months), over which there should be no uncertainty.

The increased loss to follow-up or, in general, the high percentage of missing data account for a very frequent bias in the studies meant to assess the long-term outcomes, including obesity/overweight risk at 3–6 years.

This problem concerns both prospective and retrospective studies, as it is very difficult to collect reliable data when it comes to factors seldom recorded, such as those related to diet. This difficulty produces a recall bias even when validated questionnaires and interview techniques are used.

Another possible risk of bias is related to the inclusion of studies that used regression analysis between the age(s) at the introduction of CF and the risk of overweight or obesity without comparing specific age groups, but considered age as the independent variable by which risk, i.e., the dependent variable, is assessed.

In this way, no age-specific results are obtained. As a result, only if no difference is observed by this analysis can the result be said to apply to all ages, including the ages of the PICOs in question.

By contrast, should a relationship exist between risk and age, it would not be possible to extrapolate to which age(s) the relationship in question would be related and whether it would be related to the ages of these PICOs under study, thus making it impossible to include the study.

Hence, the result turned into a criterion for including or excluding studies based on the PICOs of interest, leading to a potential selection bias, since studies with a certain result would have been included and others with a different result would have been excluded. When the studies excluded were examined through sensitivity analysis, it was found that their results were consistent with those of the studies included, thus ruling out the occurrence of a selection bias.

Hence, there is very little evidence currently available to answer the questions of this SR with any degree of confidence. Nevertheless, even in the presence of a significant bias, the results are consistent: there are no different risks of overweight/obesity or growth deficit risks related to the ages of the introduction of CF considered in this review, irrespective of the type of milk feeding. As for the effect on the iron status, exclusive breastfeeding until 6 months of age does not result in an increased risk of iron deficiency anemia or low serum ferritin levels.

One study reported a statistically significant difference in ferritin levels that are, on average, lower in EBF infants up to 6 months. However, this statistical significance did not translate into clinical significance because, in both groups, the average ferritin values fell within the normal range for the age group in question.

As for the risk of other NCDs, including DM2 and hypertension, no studies were found with a focus on the age range for the introduction of CF considered in this review.

### 4.1. Quality of Evidence

The quality of evidence of the RCTs included in this SR is not high, but moderate. The reason to downgrade the quality of evidence is that there is only one study with a small sample population for each exposure factor (EBF or EFF/mixed).

Observational studies are generally considered to be of low quality, but, as already mentioned, the consistency of the results makes it possible to be moderately confident in the evaluation of the effect.

### 4.2. Agreements and Disagreements with Other Studies or Reviews

The update provided by this SR has essentially confirmed the results of previous SRs.

### 4.3. Limitations of the SR and Potential Bias in the Review Process

A comprehensive search strategy was adopted to search and find all relevant studies.

Even the studies excluded because of poor methodological quality reported results in line with those of the studies included.

Different outcome measures were used in the studies in question thus making it not always possible to merge the results, some of which only appeared in narrative form.

Due to the small number of studies included in the analyses, the publication bias could not be assessed.

## 5. Conclusions

The analysis of scientific evidence did not reveal any significant differences in nutritional and metabolic outcomes both in the short term (growth and iron status), and in the long term (risk of overweight/obesity, DM2, and hypertension) in EBF or EFF infants starting CF at 4–6 months or 6 months.

The results of this review confirm both the WHO recommendations [1] and the EFSA scientific opinion that exclusive milk feeding is nutritionally adequate up to 6 months of age [2]. The same trust must be given to the use of the starting formulas for the first six months of life, in particular for covering the need for iron.

Therefore, it has to be very clear that introducing complementary foods before 6 months in healthy term infants EBF and EFF living in developed countries does not present any advantages.

In addition, for breast-fed infants, replacing one or two feedings with complementary food results in a lower intake of HM, hence there will be a lower intake of non-nutritional factors (for instance, immunological and neurological), which are important for optimal development in the first years of life.

Therefore, we reiterate what was stated in the EFSA SR [2]: introducing solid foods before 6 months of age is possible, but the starting of foods earlier than 6 months (180 days) is neither necessary nor desirable.

Ethics requires from all health professionals that anything that does not benefit people’s health, and in particular children, must not be recommended or practiced.

## Figures and Tables

**Figure 1 nutrients-14-00702-f001:**
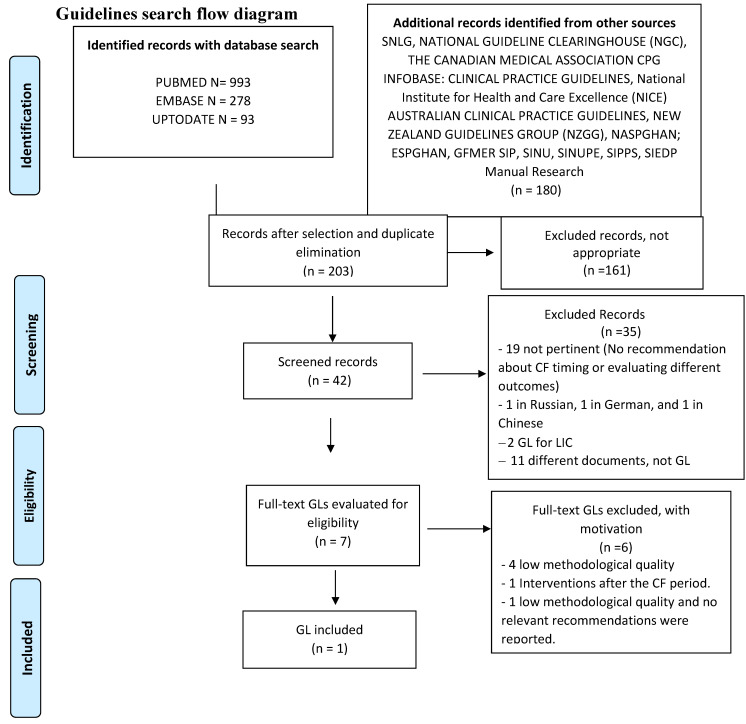
GLs search flow diagram.

**Figure 2 nutrients-14-00702-f002:**
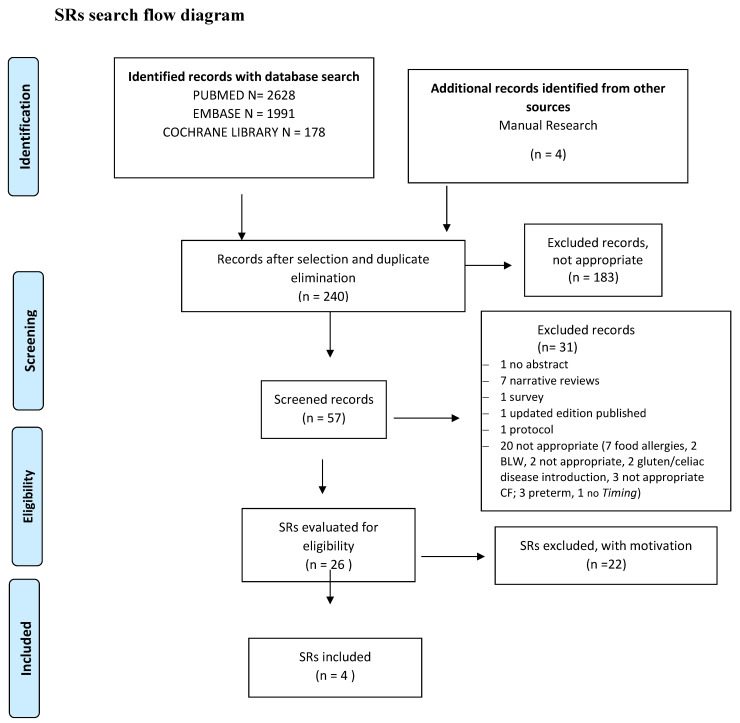
SRs search flow diagram.

**Figure 3 nutrients-14-00702-f003:**
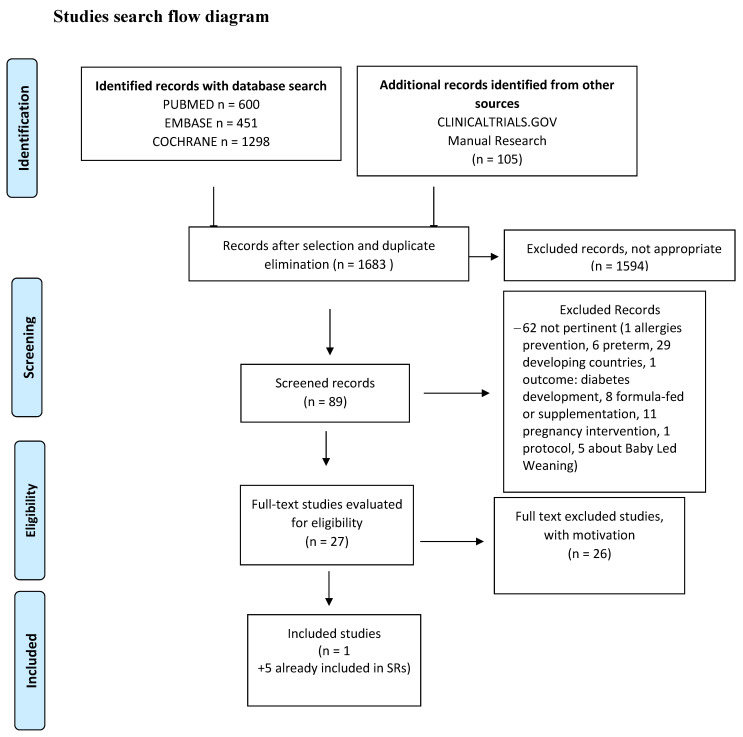
Studies search flow diagram.

## Data Availability

Not applicable.

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
