# Peer review of "Timing of Complementary Feeding, Growth, and Risk of Non-Communicable Diseases: Systematic Review and Meta-Analysis"

_nutrients, 2022, doi:10.3390/nu14030702_

Round 1

Reviewer 1 Report

The current report demonstrated the introducing Complementary Feeding (CF) before 6 months in healthy term-born infants living in developed countries from Meta-analysis of 7 studies from literature search. Please conduct the concerns below.

  1. Backgrounds of CF were not introduced in clear.
  2. It seems better to add “in infants” after feeding in the title.
  3. Cases living in Western industrialized countries were unclear. Is it possible to indicate the name of countries?
  4. It seems better to show the main data that supporting the hypothesis in figures and/or tables. Linkage with Supplementary Materials was not a good manner.
  5. On the risk of developing (later in life) type 2 diabetes mellitus (DM2) and hypertension needs to compare in detail. Particularly, the possible reason(s) shall be discussed even speculation only.
  6. The infants living in industrialized countries were used in this analysis. Is it compared with infants in developing area?
  7. In conclusion, introducing complementary foods before 6 months in healthy infants does not give any advantage. It needs a clear evidence from comparison.
  8. Limitation of the current report is required.

Author Response

Dear Reviewer,

Thank you for your comments and concerns about our paper. This helps a lot to improve it.

Here below please find our responses to your concerns.

The current report demonstrated the introducing Complementary Feeding (CF) before 6 months in healthy term-born infants living in developed countries from Meta-analysis of 7 studies from literature search. Please conduct the concerns below.

  1. Backgrounds of CF were not introduced in clear.

Thank you for your remark. We made some modifications to the paragraph in question.

2. It seems better to add “in infants” after feeding in the title.

Thank you for your comment. Actually, “complementary feeding” applies exclusively to infants, as per the definition given by WHO: “Complementary feeding is defined as the process starting when breast milk alone is no longer sufficient to meet the nutritional requirements of infants, and therefore other foods and liquids are needed, along with breast milk” (see: https://www.who.int/elena/titles/complementary_feeding/en/), so, accordingly, we think the adding is not needed. In general “in infants…” is added when the paper is addressed to some special group of infants, such as preterm born, or malnourished and so on.

3. Cases living in Western industrialized countries were unclear. Is it possible to indicate the name of countries?

Thank you for your question. The references come from several industrialized countries. However in this case is not important to detail which countries, as the main difference between industrialized or developed countries and those ones less industrialized, or still on the way of developing their economy, is the availability of foods and the rate of malnourished children under 5 years of age. In developing countries, the different availability of safe and adequate food significantly and negatively affects the nutritional status of the child in his/her first months of life. Finally, having a small weight for the gestational age at birth, which is very frequent in developing countries, constitutes a risk factor for the development of NCDs in later ages and therefore this would have been a further confounding factor for understanding the role of CF in the development of long-term NCDs.

4. It seems better to show the main data that supporting the hypothesis in figures and/or tables. Linkage with Supplementary Materials was not a good manner.

Thank you for your observation. Yes, the suggestion is more than fair. We have put tables and figures in the supplementary material so as not to make the article too long. Following the suggestion, however, we have included some of the material in the general text.

5. On the risk of developing (later in life) type 2 diabetes mellitus (DM2) and hypertension needs to compare in detail. Particularly, the possible reason(s) shall be discussed even speculation only.

Thank you for your suggestion. We did not find evidence related to these two outcomes, as reported in the Results (line 258), so no comments were included in the Discussion.

6. The infants living in industrialized countries were used in this analysis. Is it compared with infants in developing area?

Thank you for your request. Western industrialized countries only were considered in this analysis, for several reasons: first of all, and in spite of differences even inside each industrialized country due to different SES among families, the average possibilities in these countries are higher (or even much higher) than those available to families living in low-income areas of our world, hence the different offers to weaning infants are not comparable. Secondarily, cultural (and feeding practices culturally-related) use to differ more between high and low-income countries than among high-income countries alone, hence – once again – a comparison is quite difficult. Nonetheless, your objection deserves an explanation, so we added a paragraph in the discussion to clarify this point. (line 353)

7. In conclusion, introducing complementary foods before 6 months in healthy infants does not give any advantage. It needs a clear evidence from comparison.

Thank you for your remark. HM is a complete food and, in particular, a very precious health determinant for the baby.

The reduction of the HM intake, replacing 1 or more feedings with other foods, is often justified with growth needs or with the risk of iron deficiency.

The SR demonstrates that a diet with only HM in the first 6 months of life does not involve significant differences in any of the growth and metabolic outcomes considered compared, to a diet that also includes CFs. Milk is a complete food and, in particular, HM is a very precious health determinant for the baby before 6 months of age and is a complete food, nutritionally speaking. Formulas used for the first semester of life, the so-called starting formulas, have been prepared to try to be similar as possible to the nutritional composition of HM. Even though they are not equal to HM, their nutritional composition is perfectly adequate to sustain healthy growth for the first six months.

8. Limitation of the current report is required.

Thank you once again for your remark. Section 4.3 analyzes the limitations of SR.

Reviewer 2 Report

The authors summarize findings on the topic of timing of CF and communicable diseases.

Major comments

Is this a systematic review or an umbrella review (overview of SR)? The authors seem to have included systematic reviews and meta-analyses

There is no mention of the AMSTAR-2 tool for the ROB assessment in the methods, although the authors seem to have conducted the AMSTAR-2 assessment, within the supplementary files

The manuscript title is incomplete –it does not seem to reflect the review question

The “objectives” section is written in a convoluted and confusing way--- The authors have conducted a SR (or umbrella review?) and should therefore adhere to PICO criteria for defining a very clear review question. “Different short-term and long-term nutritional and metabolic outcomes” in my view, is way too broad.

The outcomes are not well defined within the study protocol, e.g., How were obesity and related NCD’s defined (through physician diagnosis? By the authors?) Were the outcomes defined expected at 3-6 years of age for all outcomes or just obesity--would one expect an outcome of hypertension and DM Type 2 at 3-6 years of age?

Lines 184-194 The assessment of Risk of bias (ROB) was not defined in the study methodology

Line 188: “data extraction, were carried out by at least two authors” which 2+ authors?

Reference 8 is not valid, the DOI does not work (unpublished?)

The prisma flow diagram should be in the manuscript and not supplementary files

What is the point of meta-analysis when there are only 1 RCT shown per outcome?

Please cite more relevant literature on the topic of CF and obesity, and NCD's

Minor comments

Abstract: “is essentially useless” is not scientific language, please adapt

“complementary feeding” should not be capitalized at first use

Lines 44: I am not aware that the World Health Organization (WHO) has recommended EBF 4- 6 months previously, and I am not sure it is relevant, it is important to state their current recommendation of 6 months EBF.

Line 81. WHO abbreviation was already defined. WHO recommends CF starts at six *completed* months

Lines 98-100 are the same as lines 101-103, please delete repeated text.

Thanks for your consideration and the opportunity to review this manuscript on this important topic.

Author Response

Dear Reviewer

Thanks a lot for your observations. We tried to respond to all your comments as below.

The authors summarize findings on the topic of timing of CF and communicable diseases.

Major comments

Is this a systematic review or an umbrella review (overview of SR)? The authors seem to have included systematic reviews and meta-analyse

We thank the reviewer for the question, which allows us to clarify.

Overviews of Reviews (Overviews) are similar to reviews of interventions, but the unit of searching, inclusion and data analysis is the systematic review rather than the primary study.

[Pollock M, Fernandes RM, Becker LA, Pieper D, Hartling L. Chapter V: Overviews of Reviews. In: Higgins JPT, Thomas J, Chandler J, Cumpston M, Li T, Page MJ, Welch VA (editors). Cochrane Handbook for Systematic Reviews of Interventions version 6.2 (updated February 2021). Cochrane, 2021. Available from www.training.cochrane.org/handbook].

This is not an overview of systematic reviews, but it is a systematic review of primary studies aimed at defining recommendations on the most appropriate age of introduction of complementary foods. The research of the studies, conducted on the main bibliographic databases (PubMed, EMBASE, Cochrane Central Register of Controlled Trials - CENTRAL), was extended to the references of the evidence-based LG, to the results of pertinent and methodologically acceptable systematic reviews, in addition to all the other sources reported in Appendix 1 (e.g. appropriate national, regional and subject-specific bibliographic databases, trials registers, etc.).

Searching the studies included in systematic reviews on the same topic is highly desirable.

 [Lefebvre C, Glanville J, Briscoe S, Littlewood A, Marshall C, Metzendorf M-I, Noel-Storr A, Rader T, Shokraneh F, Thomas J, Wieland LS. Chapter 4: Searching for and selecting studies. In: Higgins JPT, Thomas J, Chandler J, Cumpston M, Li T, Page MJ, Welch VA (editors). Cochrane Handbook for Systematic Reviews of Interventions version 6.2 (updated February 2021). Cochrane, 2021. Available from www.training.cochrane.org/handbook]

In Appendix 5 the “SUMMARY OF FINDINGS FOR THE MAIN COMPARISONS” of our SR reports only studies.

There is no mention of the AMSTAR-2 tool for the ROB assessment in the methods, although the authors seem to have conducted the AMSTAR-2 assessment, within the supplementary files

Thank you for your comment. As you rightly observed, the supplementary files contain the AMSTAR-2 assessment. Nonetheless, it is undoubtedly correct to mention this point in the methods section. We have now added this point (lines 130-133).

The manuscript title is incomplete –it does not seem to reflect the review question

Thanks for the appropriate observation, we have better defined the subject of the questions of the SR in the title that has been changed: TIMING OF COMPLEMENTARY FEEDING, GROWTH, AND RISK OF NON-COMMUNICABLE DISEASES: SYSTEMATIC REVIEW AND META-ANALYSIS.

The “objectives” section is written in a convoluted and confusing way--- The authors have conducted a SR (or umbrella review?) and should therefore adhere to PICO criteria for defining a very clear review question. “Different short-term and long-term nutritional and metabolic outcomes” in my view, is way too broad.

Thank you for your observation. "Different short-term and long-term nutritional and metabolic outcomes" is very generic indeed. For this reason, the outcomes have been specified in more detail in the following period (lines 104-124). It was also added that the questions structured as P.I.C.O. are reported in Appendix 1 (line 125)

The outcomes are not well defined within the study protocol, e.g., How were obesity and related NCD’s defined (through physician diagnosis? By the authors?) Were the outcomes defined expected at 3-6 years of age for all outcomes or just obesity--would one expect an outcome of hypertension and DM Type 2 at 3-6 years of age?

Thanks a lot for your questions. Each outcome was better defined in the SR, and, for each of them, the age of data collection was specified. (lines 104-124).

Lines 184-194 The assessment of Risk of bias (ROB) was not defined in the study methodology

Thank you for raising this issue. The general criteria of the Methodology have already been described in our previous recently published article reported in the bibliography  [Nutrients 2022, 14(2), 257; https://doi.org/10.3390/nu14020257].  As suggested by the Reviewer, we have also specified in the SR the tools used for the assessment of the studies.

Line 188: “data extraction, were carried out by at least two authors” which 2+ authors?

The data extraction work was divided between 3 groups, of 2 authors each.  The data extraction of all studies was subsequently checked by all 6 authors (M.C.V., I.S., M.B., G.S., B.C., G.T.). We have added the authors who did the data extraction in Author Contributions. (line 457)

Reference 8 is not valid, the DOI does not work (unpublished?)

You are right. At the time when we submitted the present paper, the previous one was still under review. Now it has been published, and the correct reference is: [9] “Nutrients 2022, 14(2), 257; https://doi.org/10.3390/nu14020257 ”. We have changed it in the bibliography.

The prisma flow diagram should be in the manuscript and not supplementary files

Thank you for your observation. We added the Prisma flow diagram in the manuscript.

What is the point of meta-analysis when there are only 1 RCT shown per outcome?

Thank you for your question. Although by definition, meta-analysis applies to pooled data from 2 or more studies, it is common to find meta-analyses that include only 1 study, even in SRs Cochrane, for several reasons.

  1. it is formal respect for the protocol that provides for the meta-analysis of the results, even if the systematic review led to the selection of only one study;
  2. can give immediate evidence that there is only one study on the specific outcome and no more studies that cannot be merged;
  3. calculates effect measures not reported in the original study;
  4. offers a graphical representation of the results with the forest plot.

Please cite more relevant literature on the topic of CF and obesity, and NCD's

Thank you for raising this point. We cited the recent UNICEF text in this regard

Minor comments

Abstract: “is essentially useless” is not scientific language, please adapt

Thank for the comment, however, we have found several scientific usages of this specific clause, one for all: “Lustig. Hypothalamic obesity after craniopharyngioma: mechanisms, diagnosis, and treatment. Front Endocrin 2011;2:60. doi: 10.3389/fendo.2011.00060”. Furthermore, given the importance of breastfeeding, its protective effect against various pathologies, including the development of breast cancer for the mother, and also considering the economic cost of introducing solid foods before six months, and their unnecessary introduction in the context of maintaining a good nutritional state of the infant, it seemed important to us to underline the concept with strong terms.

“complementary feeding” should not be capitalized at first use

Thank you, we corrected that.

Lines 44: I am not aware that the World Health Organization (WHO) has recommended EBF 4- 6 months previously, and I am not sure it is relevant, it is important to state their current recommendation of 6 months EBF.

Again, thanks a lot for your very accurate review. You are right, it is more important to stress the current recommendations. Nonetheless, and for the sake of truthfulness, please remember that in the past (e.g., in “Weaning: from breast milk to family food, a guide for health and community workers” [1988] - https://apps.who.int/iris/bitstream/handle/10665/39335/9241542373_eng.pdf?sequence=1&isAllowed=y ), WHO together with UNICEF recommended EBF starting time at 4-6 months of age (quotation: “2. Weaning foods. When should mixed feeding start? When a baby is about 4-6 months old, the mouth starts to become ready to accept non-liquid foods. […] Clearly, during this time children are becoming ready to eat some solid food”), that’s why we stressed this point. And despite the current WHO recommendations, many colleagues in several countries are still bound to that ancient indication and suggest families start EBF beforehand, i.e., before 6 months of age.

Line 81. WHO abbreviation was already defined. WHO recommends CF starts at six *completed* months

Thank you for the thorough review which we greatly appreciated. We corrected that mistake (double definition of WHO). As for your other objection at this line, we would like to stress that either our expression or your clarification might be misleading: our definition might make somebody think that 6 months means in the 6th month, while “six completed months” might be interpreted as the ending days of the month six. Just to avoid any misunderstanding on this crucial point, we added “180 days of life”. We hope you agree with this decision.

Lines 98-100 are the same as lines 101-103, please delete repeated text.

The phrases are not repeated, as in the first group of lines you quoted, the comparison is made “with exclusive breastfeeding up to 6 months”, whereas in the second group the comparison is made “with exclusive formula or mixed feeding (human milk + baby formula) up to 6 months”

Thanks for your consideration and the opportunity to review this manuscript on this important topic.

It is we who thank you for all the comments made, which testify to a remarkable competence in the field and a very careful reading of our text. This honors and gratifies us.